# Fostering active choice to empower behavioral change to reduce cardiovascular risk: A web-based randomized controlled trial

Lorraine L. Landais[1], Judith G. M. Jelsma[1], Olga C. Damman[1], Evert A. L. M. Verhagen[1,2], Danielle R. M. Timmermans[1]*

1 Department of Public and Occupational Health, Amsterdam UMC, Amsterdam Public Health Research institute, Vrije Universiteit Amsterdam, Amsterdam, The Netherlands, 2 Amsterdam Collaboration on Health & Safety in Sports, Department of Public and Occupational Health, Amsterdam Movement Sciences, Amsterdam UMC, Vrije Universiteit Amsterdam, Amsterdam, The Netherlands

* drm.timmermans@amsterdamumc.nl

**Data Availability Statement:** All relevant data are within the manuscript and its Supporting Information files. Additional file 1 presents the code used for the statistical analyses.

## Abstract

### Objective

To investigate the effect of an active choice (AC) intervention based on creating risk and choice awareness–versus a passive choice (PC) control group–on intentions and commitment to cardiovascular disease (CVD) risk-reducing behavior.

### Methods

Adults aged 50–70 (n = 743) without CVD history participated in this web-based randomized controlled trial. The AC intervention included presentation of a hypothetical CVD risk in a heart age format, information about CVD risk and choice options, and a values clarification exercise. The PC group received a hypothetical absolute numerical CVD risk and brief information and advice about lifestyle and medication. Key outcomes were reported degree of active choice, intention strength, and commitment to adopt risk-reducing behavior.

### Results

More AC compared to PC participants opted for lifestyle change (OR = 2.86, 95% CI:1.51;5.44), or lifestyle change and medication use (OR = 2.78, 95%CI:1.42;5.46), than 'no change'. No differences were found for intention strength. AC participants made a more active choice than PC participants (β = 0.09, 95%CI:0.01;0.16), which was sequentially mediated by cognitive risk perception and negative affect. AC participants also reported higher commitment to CVD risk-reducing behavior (β = 0.32, 95%CI:0.04;0.60), mediated by reported degree of active choice.

### Conclusions

Fostering active choices increased intentions and commitment towards CVD risk-reducing behavior. Increased cognitive risk perception and negative affect were shown to mediate the effect of the intervention on degree of active choice, which in turn mediated the effect on

**Funding:** Funded by Amsterdam University Medical Center. The funders had no role in study design, data collection and analysis, decision to publish, or preparation of the manuscript.

**Competing interests:** The authors have declared that no competing interests exists

commitment. Future research should determine whether fostering active choice also improves risk-reducing behaviors in individuals at increased CVD risk in real-life settings.

## Trial registration

ClinicalTrials.gov: NCT05142280. Prospectively registered.

## Introduction

Cardiovascular diseases (CVDs) are the leading cause of death worldwide [1]. In primary CVD prevention, individuals with moderate to high CVD risk are typically recommended to make lifestyle changes, including smoking cessation, a healthy diet, sufficient physical activity, and, if needed, use of preventive medication [2, 3]. However, many individuals at increased CVD risk struggle to initiate and maintain lifestyle changes and adhere to preventive medication [4–6]. Adherence has mainly been studied with medication, but it has also become a research topic in lifestyle behaviors [7, 8]. Nonadherence can be intentional, i.e. when individuals decide not to follow recommendations based on their own beliefs, priorities, preferences, or experiences (e.g., side effects), or unintentional, i.e. when individuals lack the capacities or resources to follow recommendations (e.g., memory deficiencies) [9, 10]. Poor adherence to healthy lifestyle and medication regimen are associated with poor clinical outcomes at the individual level and with increased healthcare costs at the societal level [11, 12].

In addition to individual factors, ineffective communication between healthcare professionals and patients might contribute to nonadherence [13–15]. In general, communication in primary CVD prevention seems to be focused on advice-giving without active engagement in decision-making on the part of patients themselves [3, 16]. More active choices may be needed to prevent intentional and unintentional nonadherence. Previous studies concluded that individuals are often inadequately informed about pros and cons of CVD prevention choice options and that their personal preferences and values are insufficiently considered [13, 15]. Consequently, professionals' advices may be discordant with patients' priorities [17]. Ineffective communication can also negatively impact patient understanding of the importance of behavioral change and adherence [18, 19]. In contrast to active choices, passive choices can be characterized by little individual autonomy and lack of choice awareness [20]. If healthcare professionals would support individuals in making a more active choice regarding their CVD risk, individuals' choices may better align with their values, fostering commitment to CVD risk-reducing behavior. As such, active choices may decrease both intentional nonadherence (e.g., because individuals actively consider their values and preferences) and unintentional nonadherence (e.g., because individuals become aware of factors that otherwise would have remained unknown).

We defined an active choice regarding coping with increased CVD risk as a conscious and autonomous choice in which an individual (a) understands his/her CVD risk; (b) is aware of the different options to cope with the risk, understands what options mean, and actively evaluates pros and cons of the options; and (c) thinks about personal values and preferences in relation to the choice. This definition is based on the stages distinguished in an analytic decision-making style, which include: awareness of the choice problem, structuring the choice problem, evaluating the choice problem, and integrating evaluations into a choice [21, 22]. Our definition is also informed by Self-Determination Theory (SDT), which posits that autonomy (i.e., a sense of choice about one's behavior) is a basic psychological need, in addition to competence

and relatedness [23]. Satisfaction of those needs predicts autonomous motivation [24, 25] which has been positively associated with sustained behavior change and medication adherence [23, 26–28]. Furthermore, perceived autonomy in decision-making has been associated with intrinsic motivation, satisfaction, well-being, and behavioral persistence [24, 29]. A growing body of evidence shows that patients who make active and autonomous health decisions have better healthcare experiences and outcomes [30].

As the first step of an active choice process for CVD risk reduction, it is essential that individuals are aware of their CVD risk and perceive a discrepancy between their high-risk situation and the desired lower-risk situation [21, 31, 32]. How CVD risk is presented influences individuals' cognitive and affective risk perception [33, 34]. In particular, the presentation of CVD risk as 'heart age', seems relevant to making high-risk individuals aware of such discrepancy. Although there is some debate about the use of heart age because it may, for example, inflate affective risk appraisals [35, 36], this format has been shown to lead to higher risk perception compared to absolute numerical risk formats [2], and in some studies to more accurate risk perception [37], or improved risk understanding [34]. According to Loewenstein et al. [38], the relationship between an individual's cognitive risk perception on his or her behavioral response is, at least in part, mediated by affect. In addition to making people aware of the size of their disease risk, it is also considered important to inform them about the risk factors, symptoms, causes, and consequences related to CVD risk, since it is known that lay representations of disease and disease risk are sometimes discordant with scientific insights [31, 39, 40]. For medication use, individuals' perceptions about the benefits and harms of treatment (i.e. necessity beliefs and concerns) are important; adherence is more likely if necessity beliefs outweigh the concerns [40].

To make an active choice, it is further important for individuals to become aware of available choice options, understand those options, and weigh the associated pros and cons. In shared decision-making, so-called 'patient decision aids' are evidence-based tools specifically developed for this purpose. These tools describe the health condition or problem, make the decision to be made explicit, provide an overview of options with pros and cons, and help individuals clarify values and preferences [41, 42]. Compared to usual care, use of decision aids makes individuals feel more knowledgeable, better informed, and clearer about their values [41]. Clarifying values and preferences specifically enables individuals to make choices consistent with their values and preferences [43]. This has been suggested to increase commitment towards chosen courses of action [44, 45]. Commitment (i.e., attachment to or determination to reach a goal) has been associated with greater behavioral persistence [46, 47].

The current study aimed to investigate the effect of an active choice intervention, based on increasing risk perception and choice awareness, on intentions and commitment to CVD risk-reducing behavior, compared to a passive choice condition based on information and advice only, using hypothetical scenarios. We randomized participants to an experimental condition that either promoted an active choice (intervention) or passive choice (control). The presented information about CVD risk and risk-reducing options was expected to improve participants' knowledge and perceived efficacy of lifestyle changes and medication use in reducing CVD risk (i.e., response efficacy). Based on the previously described literature, we expected that the active choice intervention would lead to a higher reported degree of active choice and autonomous motivation, higher intention strength, and more commitment towards the chosen option (i.e., lifestyle change, medication use, or both). Intention strength and commitment were used as a proxy for behavioral change. We had no prior expectations regarding between-group differences in the type of option participants would select to cope with CVD risk since all options, including no behavioral change, could either actively or passively be chosen [26]. We additionally performed mediation analyses to investigate (a) whether the heart age

presented in the active choice intervention increased participants' risk awareness (i.e., cognitive and affective risk perception), contributing to a more active choice process; and (b) whether reported degree of active choice and autonomous motivation mediated the relationship between experimental condition (intervention/control) and commitment. This proof of concept study aimed to test the preliminary efficacy of fostering an active choice of participants at increased cardiovascular risk for preventive action as well as to better understand the mediating psychological processes using hypothetical scenarios. If our study shows positive results, the intervention could be tested in a real-life trial.

## Methods

### Transparency and openness

Following the CONSORT guideline for reporting parallel group randomized trials, this article reports data exclusions, manipulations, measures included, and sample size determination. Additional file 1 presents the analysis code. Data were analyzed using SPSS for Windows version 26 (IBM Corp.). The study was prospectively registered on clinicaltrials.gov (NCT05142280).

### Design and setting

We employed a two-arm parallel study design and used hypothetical CVD risk scenarios for feasibility reasons. The target group consisted of relatively healthy adults. We targeted people who might be confronted with increased CVD risk in the future as opposed to people already at increased risk and people who likely will remain at low risk. Eligible individuals were randomly assigned to an experimental web-based active choice (AC) intervention or passive choice (PC) control condition. Both groups were presented with a hypothetical scenario and asked to imagine that they were at increased CVD risk. The Medical Ethics Review Committee of VU University Medical Center confirmed that no ethical approval was required for this study (2021.0676).

### Participants

Dutch adults aged 50–70 years were recruited within the online ISO-certified StemPunt panel of research agency Motivaction. Panel members voluntarily signed up and earned points for participation, to be exchanged for gift cards. Smokers, members using blood- or cholesterol lowering medication, members with CVD, a history of CVD, diabetes, kidney damage, or rheumatism were not eligible because these factors are related to CVD risk. Pregnant women, wheelchair users, and individuals unable to walk a minimum of 100 meters were excluded. Finally, we excluded individuals who rated their diet and/or physical activity levels as 'excellent' (5-point scale from 'bad' to 'excellent') since we aimed to obtain a sample who are likely confronted with increased CVD risk in the future and who may benefit from lifestyle improvements.

### Active choice intervention

The AC intervention aimed to promote an active choice regarding lifestyle changes and medication use to lower CVD risk. Intervention components were based on the decision-making literature, specifically about stages in analytic decision-making [21], and quality criteria for patient decision aids [48]. Furthermore, the content and framing of information presented in the intervention was carefully composed because this is known to affect decision-making

[49, 50]. The intervention was pre-tested by six relatively healthy middle-aged adults before data collection; subsequently, some adaptations were made to increase comprehensibility.

AC participants read a vignette in which they were asked to imagine they were in consultation with their general practitioner (GP). The GP explained that the participant had high blood pressure and that based on his or her blood pressure, cholesterol, age, and gender, he or she had a 'heart age' of 13 years above the actual age, meaning that he or she was at increased risk of dying from CVD. After this imaginary risk, participants were informed about symptoms of high blood pressure and potential causes and consequences of increased CVD risk [31].

Usually, a heart age score is calculated by comparing an individual's current absolute risk to the age at which he or she would reach that absolute risk if he or she had 'ideal' risk factors [51]. The heart age used in this study was hypothetical and not based on individual participant data. The hypothetical heart age (+13 years) was obtained by averaging increases in heart age for men and women that followed from the heart age calculator of the Dutch Heart Foundation [52] using the following high-risk values: 65 years old, no smoking, systolic blood pressure: 160 mm Hg, total cholesterol: 7 mmol/L, HDL cholesterol: 1 mmol/L.

The active choice intervention showed four options to cope with increased CVD risk: (1) lifestyle change; (2) medication use; (3) lifestyle change and medication use; or (4) no change. Each option was subsequently explained on a different webpage, including (1) the option's meaning; (2) the pros; (3) the cons. For the first three options, the pros included gain-framed messages about risk reductions (i.e., focused on benefits of the options). Risk reductions were presented as reductions in years that corresponded with the initial heart age of +13 years: a 4-year reduction for diet, a 2-year reduction for physical activity, and a 3-year reduction for blood pressure-lowering medication. We calculated the reductions using the relative risk reductions reported in a previous study [53]. A specific note informed participants that heart age reductions were general estimates and that reductions in years (e.g., for diet/physical activity) could not simply be added because of their interrelatedness. For each option, participants were asked to select one or more pros or cons considered most important.

Thirdly, the intervention provided a values clarification exercise asking participants to indicate to what extent they agreed with six statements on a scale of 1 (totally disagree) to 10 (totally agree). The statements included values associated with lifestyle change and medication use, including the desire to live in good health as long as possible and the desire not to get foreign substances in one's body. To explicitly show the implications of participants' values, they were presented with the choice option that corresponded with their rating on each statement (i.e., lifestyle change or medication use). Additional file 2 presents an English translation of the AC intervention.

### Passive choice control condition

The passive choice (PC) control condition was based on usual care in Dutch GP practice and was thought to promote a more passive choice regarding how to cope with CVD risk. Usual care was analyzed by checking Dutch GP guidelines for CVD risk management [54] and interviewing two Dutch GPs about their communication with patients about CVD risk. The PC condition started with a vignette in which participants were asked to imagine that they were in consultation with their GP. In this condition, increased CVD risk was presented as an absolute numerical risk [55]: an 8% risk of dying from CVD within 10 years. This corresponded to a heart age of +13 years since it was calculated using the same risk values as for the AC condition. Participants were informed that the presented risk was hypothetical. Subsequently, they read a hypothetical GP advice to live a healthy lifestyle and take medication to lower their

CVD risk followed by brief information about lifestyle changes and medication use. Additional File 3 presents the passive choice condition.

## Procedure

The research agency invited panel members (N = 12,593) by e-mail. Interested panel members were asked nine questions to check eligibility. The research agency randomly assigned eligible individuals (N = 917) to the AC or PC condition (1:1 ratio) using XS5 software. Randomization was stratified by gender. Participants were blind to the content presented to the other group. Data collection took place between January 19 and February 8, 2022. All included participants provided informed consent.

## Outcomes

Table 1 shows the measurements, all relying on self-report. Before data collection, the questionnaire was also pre-tested among the six adults. Based on the pre-test, we made several adaptations to the questionnaires to increase comprehensibility. Participants' intention regarding coping with CVD risk was assessed by one item in which participants could choose lifestyle change, medication use, both, or 'no change'. Those who intended to change their lifestyle were also asked which behavior they would want to change. Autonomous motivation was measured using the six-item autonomous motivation subscale of the Treatment Self-Regulation Questionnaire (TSRQ), based on Self-Determination Theory [56]. The TSRQ has shown reasonable validity and reliability regarding lifestyle behaviors and medication adherence [57, 58]. The autonomous motivation subscale exhibited good internal consistency ($\alpha$ = 0.93). Cognitive risk perception (i.e., perceived likelihood of developing CVD within 10 years) was assessed by one item (5-point scale) adapted from Fair et al. [59]. Three items, adapted from Damman et al. [34], assessed participants' negative affect (i.e., feeling afraid, anxious, and worried) associated with the presented CVD risk on a 10-point scale ($\alpha$ = 0.94). Knowledge about CVD risk was measured by four self-constructed items, representing four cognitive lay illness representations: beliefs about symptoms, causes, consequences, and control of CVD risk [60]. We also measured response efficacy (i.e., the belief that carrying out the recommended action will reduce CVD risk) and self-efficacy (i.e., the belief in one's ability to perform the recommended coursed of action successfully), which together determine an individual's coping response [61]. Participants answered two self-efficacy items: one about lifestyle change and one about medication use; however, only the item with the highest score was used in the analyses. This was done because we were interested in the level of confidence participants had regarding the choice option they were most inclined to pursue.

The primary outcomes were the reported degree of active choice, intention strength, and commitment to CVD risk reducing behavior. To measure these outcomes, we used self-constructed items. These items were thoroughly evaluated with participants during the pretest and subsequently adjusted based on their experiences and feedback. The reported degree of active choice was measured using 11 self-constructed items (5-point scale). As there is no measure for active choice, we first defined the concept based on relevant literature (see introduction). We used this definition to develop our measure, as a solid theoretical foundation is essential for these kind of measures (see [62]). This scale was largely comparable to an active choice scale that we developed for a previous study [63]. In accordance with the previous study, the first items of our active choice scale were adapted from the 'Informed subscale' and 'Values clarity subscale' of the validated Decisional Conflict Scale [64]. The other subscales of the Decisional Conflict Scale were not included as they measured different constructs. Based on that study, the scale yields a reasonably normal data distribution. In the current study, we pre-

**Table 1. Outcome measures and their assessment.**

| Outcome measure | Item(s) | Scale / Response categories | Scale's Cronbach's alpha [a] | Mean (SD) | Range |
|---|---|---|---|---|---|
| Intention regarding coping with CVD risk | Initially, we showed you *[a heart age of 13 years older than your actual age / a risk of cardiovascular disease of 8%]*. Imagine that this CVD risk would really belong to you. What would you choose? | ■ Lifestyle change | - | - | - |
| | | ■ Medication use | | | |
| | | ■ Lifestyle change and medication use | | | |
| | | ■ No change | | | |
| Intended kind of lifestyle change[b] | You have indicated that you would change your lifestyle. What would you change? You may select multiple answers. | ■ A healthier diet | - | - | - |
| | | ■ More physical activity | | | |
| | | ■ Other (please specify): . . . | | | |
| Autonomous motivation [b] | The reason I would choose [chosen behaviour] is: | 1 (totally disagree) - 7 (totally agree) | 0.93 | 6.12 (0.76) | 1.83–7.00 |
| | ■ Because *[chosen behaviour]* is very important for being as healthy as possible | | | | |
| | ■ Because I personally believe it is the best thing for my health | | | | |
| | ■ Because I feel that I want to take responsibility for my own health | | | | |
| | ■ Because *[chosen behaviour]* is an important choice I really want to make | | | | |
| | ■ Because I have carefully thought about it and believe *[chosen behaviour]* is very important for many aspects of my life | | | | |
| | ■ Because [chosen *behaviour]* is consistent with my life goals | | | | |
| Cognitive risk perception | If the CVD risk would really belong to me, I think my risk of developing cardiovascular disease within 10 years is: | ■ 1 (very low) - 5 (very high) | - | 3.03 (0.85) | 1–5 |
| Negative affect | How would you feel if the CVD risk would really belong to you? | ■ 1 (not at all) - 10 (very much) | 0.94 | 6.08 (2.10) | 1–10 |
| | ■ Afraid | | | | |
| | ■ Anxious | | | | |
| | ■ Worried | | | | |
| Knowledge | ■ You usually don't notice it when you have an increased risk of cardiovascular disease | ■ Agree | - | 3.34 (0.88) | 0–4 |
| | ■ The risk of cardiovascular disease increases with age | ■ Disagree | | | |
| | ■ An increased risk of cardiovascular disease may lead to a stroke | ■ Don't know | | | |
| | ■ You can not affect the risk of cardiovascular disease yourself | | | | |
| Response efficacy | Imagine again that the CVD risk would really belong to you. | 1 (not at all) - 10 (very much) | | | |
| | ■ I believe that a healthy lifestyle would reduce my risk of cardiovascular disease | | | 8.36 (1.26) | 1–10 |
| | ■ I believe that taking blood pressure-lowering medication would reduce my risk of cardiovascular disease | | | 7.13 (1.74) | 1–10 |
| Self-efficacy [b] | Imagine again that the CVD risk would really belong to you. | 1 (not at all) - 10 (very much) | - | 8.51 (0.96) | 6–10 |
| | ■ I would be confident that I can maintain a healthy lifestyle [c] | | | | |
| | ■ I would be confident that I can maintain medication use [c] | | | | |
| Active choice | In making this choice [about coping with an increased risk of CVD], I have. . . | 1 (totally disagree) - 5 (totally agree) | 0.83 | 3.84 (0.51) | 2–5 |
| | ■ . . . taken into account the advantages of a healthier lifestyle | | | | |
| | ■ . . . taken into account the disadvantages of a healthier lifestyle | | | | |
| | ■ . . . taken into account the advantages of medication | | | | |
| | ■ . . . taken into account the disadvantages/side effects of medication | | | | |

*(Continued)*

**Table 1.** (Continued)

| Outcome measure | Item(s) | Scale / Response categories | Scale's Cronbach's alpha [a] | Mean (SD) | Range |
|---|---|---|---|---|---|
| | ▪ . . . thought about the advantages and disadvantages that would be most important to me | | | | |
| | ▪ . . . thought about what a healthier lifestyle would do to my risk of cardiovascular disease | | | | |
| | ▪ . . . thought about what medication would do to my risk of cardiovascular disease | | | | |
| | ▪ . . . thought about the time, effort and energy it would take to change my lifestyle | | | | |
| | ▪ . . . thought about the effort it would take to use medication | | | | |
| | ▪ . . . thought about what is important to me | | | | |
| | ▪ . . . thought about how important my health is to me | | | | |
| Intention strength [b] | Imagine again that the CVD risk would really belong to you. | 1 (no strong plan at all) - 10 (very strong plan) | - | 8.27 (1.06) | 5–10 |
| | ▪ How strong is your plan to change your lifestyle? [c] | | | | |
| | ▪ How strong is your plan to use medication? [c] | | | | |
| Commitment [b] | Imagine again that the CVD risk would really belong to you. | 1 (not at all) - 10 (very much) | 0.88 | 8.29 (1.00) | 4.5–10.0 |
| | ▪ I would be willing to invest time, effort and energy into *[chosen behaviour]* | | | | |
| | ▪ I would do everything I can to maintain a healthier lifestyle [c] | | | | |
| | ▪ I would do everything I can to keep taking medication [c] | | | | |

Note. Abbreviations: *CVD* cardiovascular disease

[a] Scales were constructed by averaging the responses to the total number of items

[b] Only asked participants who intended to change their behaviour (i.e., lifestyle change, medication use, or both).

[c] Participants' highest score on the two items concerning lifestyle change and medication use was used in the statistical analyses

tested the intervention and the questionnaire before data collection. One of the researchers (LL) used the 'think-aloud method': six adults went through the intervention and the questionnaire while reading everything aloud and verbalizing their thoughts aloud. This pre-test made some adjustments to the active choice scale, including clarifying the wording of some items. Our current active choice scale differs from the previous active choice scale regarding the wording of items: we adjusted the items to the current decision situation. Moreover, we added two items to the current scale to measure the extent to which participants had considered the consequences of (1) adopting a healthy lifestyle and (2) using medication for their risk of cardiovascular disease. Our active choice scale showed good internal consistency ($\alpha$ = 0.83).

Intention strength and commitment were only measured in participants who intended to make lifestyle changes and/or use medication. Two items measured intention strength regarding lifestyle change and medication use (10-point scale). Concerning commitment to adopt risk-reducing behavior, two items measured individuals' commitment to maintaining behavioural changes on a 10-point scale: one about lifestyle change and one about medication use. In the analyses, we used the highest scores on these items since we were interested in the intention strength and commitment regarding the choice option participants were most inclined to pursue. For commitment, a composite score was calculated by averaging the highest score on the two aforementioned items with the score on an additional item about the willingness to invest time, effort and energy into the chosen behavior.

## Sociodemographic variables

With participants' consent, the research agency shared information about participants' age, gender, educational level, ethnic background (i.e., non-Dutch ethnic origin meant that the individual was born abroad and at least one of the parents was born abroad [65]), and whether they had children. Before the intervention, we assessed participants' perceived health status using the first item of the RAND-36 scale (5-point scale) [66]. Participants' perceived lifestyle, nutrition and physical activity were assessed by self-constructed items that used the same wording and scale as those that assessed perceived health status.

**Process evaluation.** We conducted a process evaluation to assess the intervention's acceptability and to identify contextual factors potentially influencing outcomes. The first items addressed the extent to which the intervention was understandable, readable, well-organized, and interesting, on a scale of 1 (totally disagree) to 5 (totally agree). Items also assessed whether participants learned something from the information and if information was too much. Furthermore, participants had to rate to which extent the active choice intervention or control condition components supported them in decision-making (scale of 1 (very poorly) to 5 (very well)). A final open-ended question assessed suggestions for improvement.

## Sample size and power calculation

Since a previous study using an active choice intervention in a different setting [63] found that gender modified the effect on multiple outcomes, we stratified the current study by gender. We used SD = 0.478 for men and SD = 0.490 for women for the power calculation on 'Active choice' [63]. We used a statistical power of 80%, an alpha level of 0.05, and a clinically relevant difference of 0.145 points on 'Active choice', meaning that 173 men and 182 women were needed per group. We assumed a delta of 0.145 for active choice because this lies exactly between 0.13 and 0.16; these were the significant differences found in an earlier study [63]. Consequently, we could detect 0.74 points difference for men and 0.64 points difference for women on 'Intention strength', and 0.45 points difference for both men and women on 'Commitment'.

## Data analysis

**Main analyses** Linear regression analyses examined between-group differences in reported degree of active choice, intention strength, commitment, response efficacy, self-efficacy, autonomous motivation, cognitive risk perception, and negative affect. Heavily skewed outcomes were log-transformed to meet normal distribution assumptions. Logistic regression analyses examined between-group differences in intention regarding coping with CVD risk (multinomial logistic regression analysis), intended kind of lifestyle change, and knowledge (recoded as a variable representing the number of correct answers; analyzed by ordinal regression analysis). A pairwise deletion was performed for missing data. A significance level of .05 was used.

**Mediation analyses.** Mediation analyses were performed in SPSS using the PROCESS macro version 4.1 from Hayes [67]. The first analysis examined whether cognitive risk perception and negative affect sequentially mediated the relationship between experimental condition (i.e., AC versus PC condition) and reported degree of active choice using serial mediation (Hayes' Model No. 6). The second, parallel mediation analysis used simple mediation (Hayes' Model No. 4) to assess whether the relationship between experimental condition and commitment was mediated by (a) the degree of active choice and (b) autonomous motivation. Linear regression analyses with bootstrapping based on 5000 samples were performed to generate 95% confidence intervals for the mediating effects. A significance level of .05 was used.

**Analyses of confounding and effect modification.** We examined confounding by age on the three primary outcomes. Furthermore, we examined effect modification of gender (men/women), health condition (yes/no), and educational level (lower/medium/higher) on those three outcomes, using a significance level of .10 [68]. We additionally examined confounding and effect modification on the remaining outcomes in case of confounding or effect modification on the primary outcomes.

# Results

## Study population

Of 12,593 invited panel members, 2,378 completed the eligibility questions. Eventually, 1,454 panel members did not meet inclusion criteria, and 7 quit during the assessment. Eligible panel members (n = 917) were randomly assigned to the AC intervention or PC control condition. Next, in total 174 panel members were excluded, either due to information that was initially incorrectly presented by the research agency in the online system (n = 94), a too short completion time (i.e., <20 seconds) (n = 20), or because they dropped out (n = 60). Finally, 367 participants were included in the AC group; 376 in the PC group. Fig 1 displays the flow diagram.

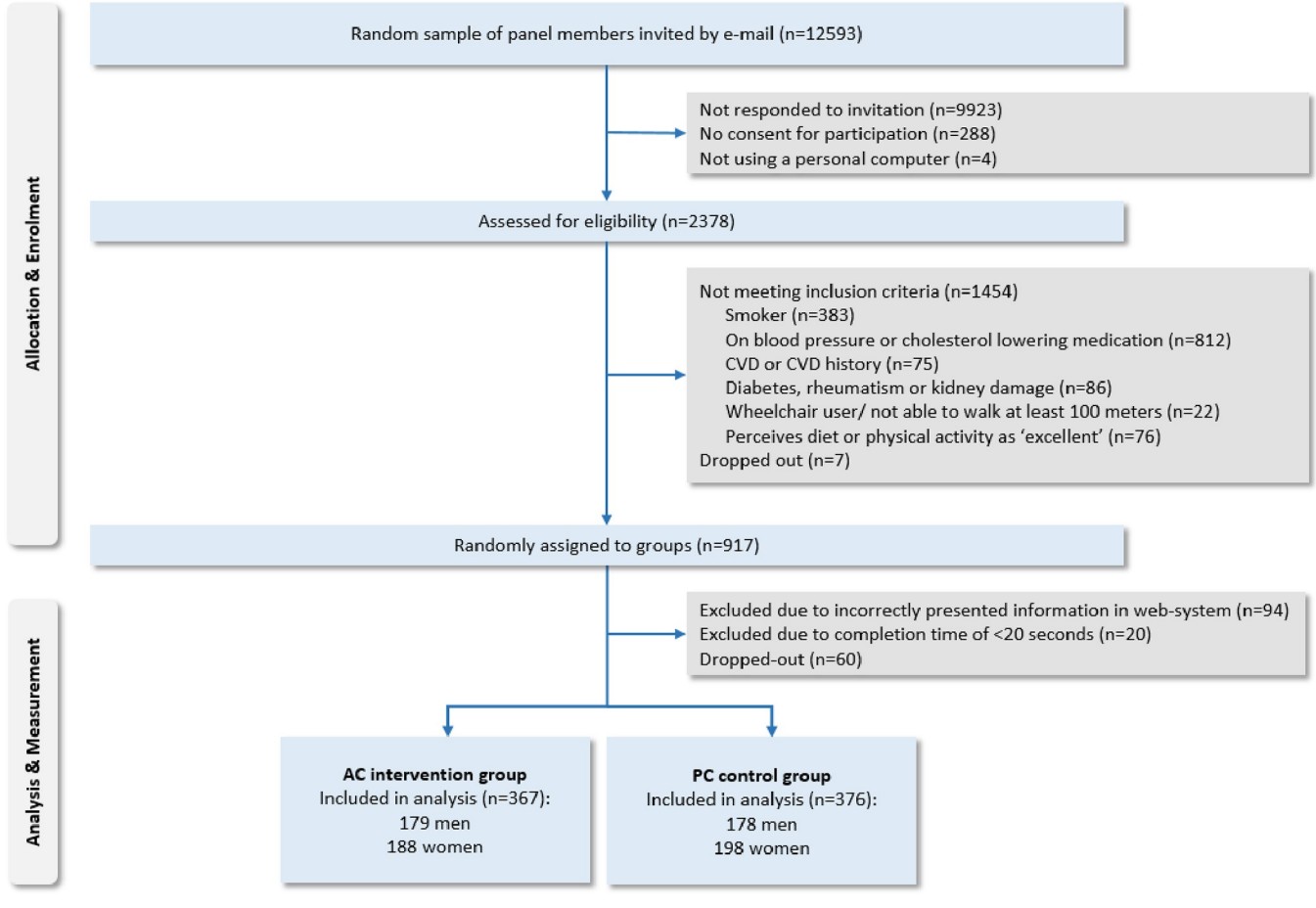

**Fig 1. Participant flow diagram.**

Table 2. Demographic characteristics at baseline.

| Demographics | Active choice (AC) group | Passive choice (PC) group | Total |
|---|---|---|---|
| | (*n* = 367) | (*n* = 376) | (*n* = 743) |
| Age (years), mean ± SD | 60.7 ± 6.0 | 60.1 ± 6.1 | 60.4 ± 6.0 |
| Gender, women, *n* (%) | 188 (51.2%) | 198 (52.7%) | 386 (52.0%) |
| Educational level, *n* (%) | | | |
| Lower | 24 (6.5%) | 30 (8.0%) | 54 (7.3%) |
| Medium | 182 (49.6%) | 197 (52.4%) | 379 (51.0%) |
| Higher | 161 (43.9%) | 149 (39.6%) | 310 (41.7%) |
| Dutch background, *n* (%) | 355 (96.7%) | 358 (95.2%) | 713 (96.0%) |
| Has children, *n* (%) | 278 (75.7%) | 272 (72.3%) | 550 (74.0%) |
| Physical or mental health condition, *n* (%) | 145 (39.8%) | 155 (42.1%) | 300 (41.0%) |
| Perceived health status [a], mean ± SD | 3.05 (0.64) | 3.01 (0.66) | 3.03 (0.65) |
| Perceived lifestyle [a], mean ± SD | 2.97 (0.56) | 2.98 (0.58) | 2.98 (0.57) |
| Perceived nutrition [a,b], mean ± SD | 3.14 (0.58) | 3.10 (0.54) | 3.12 (0.56) |
| Perceived physical activity [a,b], mean ± SD | 2.84 (0.77) | 2.90 (0.72) | 2.87 (0.74) |

Note. Abbreviations: *SD* standard deviation

[a] A 5-point scale was used: (1) Poor; (2) Fair; (3) Good; (4) Very good; (5) Excellent

[b] Participants with a score of 5 (Excellent) were excluded

Demographic information is presented in Table 2. A total of 52% of the population was female. Participants had a mean age of 60.4 years (SD = 6.0); 96% had a Dutch background, 74% had children, and 41% reported a physical or mental health condition. Lower educated participants were underrepresented (7.3%) compared to medium education (51.0%) and higher education (41.7%). Participants perceived their health status, lifestyle, nutrition and physical activity as 'good'.

**Analyses of confounding and effect modification.** No confounding by age, nor effect modification by gender, health condition, or educational level was found in the relation between the experimental condition and the three primary outcomes. Consequently, no corrections or stratifications were done.

**Results main analyses.** Table 3 presents the results of the AC interventions' main effects. For intention regarding coping with CVD risk, the results were compared both within and between study arms. Significantly more AC participants intended to change their lifestyle (OR = 2.86, 95%CI:1.51;5.44) or change their lifestyle and use medication (OR = 2.78, 95% CI:1.42;5.46) than to change nothing compared to PC participants. Furthermore, a between-group analysis comparing those who intended to change lifestyle and/or use medication (i.e., the first three categories) with those who did not intend to change (i.e., the last category) revealed that significantly fewer AC participants intended not to change compared to PC participants (OR = 0.36, 95%CI:0.19;0.68). Among participants who intended to change lifestyle (n = 670), no between-group differences were found regarding the intended kind of lifestyle change (i.e., changing diet and/or physical activity).

Autonomous motivation–an outcome that was log-transformed because of a skewed distribution–did not significantly differ between groups; β = 0.00, 95%CI:-0.02;0.02. Cognitive risk perception and negative affect were significantly higher in AC than in PC participants; β = 0.51, 95%CI:0.40;0.63 and β = 0.56, 95%CI:0.26;0.86, respectively. Concerning knowledge, AC participants were significantly more likely to correctly answer the questions about CVD risk-related symptoms, causes, and consequences than PC participants (OR = 1.47, 95%CI:

**Table 3. Results of the regression analyses comparing the effects of the active choice (AC) intervention to the passive choice (PC) control condition on the study outcomes.**

| Outcome | Outcome specification | Active Choice (AC) group | Passive Choice (PC) group | |
|---|---|---|---|---|
| | | Mean (SD) or N (%) | Mean (SD) or N (%) | β or OR [95% CI] |
| Intention regarding coping with CVD risk (n = 743) | Lifestyle change | 246 (67.0%) | 227 (60.4%) | OR = 2.86** [1.51; 5.44][a] |
| | Medication use | 6 (1.6%) | 16 (4.3%) | OR = 0.99 [0.32; 3.04][a] |
| | Lifestyle change & medication use | 101 (27.5%) | 96 (25.5%) | OR = 2.78** [1.42; 5.46][a] |
| | No change | 14 (3.8%) | 37 (9.8%) | |
| Intended kind of lifestyle change (n = 670) | Change diet | 263 (75.8%) | 235 (72.8%) | OR = 1.17 [0.83; 1.66] |
| | Change physical activity | 279 (80.4%) | 255 (78.9%) | OR = 1.09 [0.75; 1.59] |
| Autonomous motivation (n = 692) | | 6.13 (0.79) | 6.11 (0.72) | β = 0.00 [-0.02; 0.02][b] |
| Cognitive risk perception (n = 743) | | 3.29 (0.84) | 2.77 (0.79) | β = 0.51** [0.40; 0.63] |
| Negative affect (n = 743) | | 6.36 (2.17) | 5.80 (2.00) | β = 0.56** [0.26; 0.86] |
| Knowledge (n = 743) | | 3.42 (0.84) | 3.26 (0.91) | OR = 1.47** [1.11; 1.94][c] |
| Response efficacy (n = 743) | Lifestyle change | 8.48 (1.28) | 8.24 (1.23) | β = 0.24* [0.06; 0.42] |
| | Medication use | 7.16 (1.78) | 7.10 (1.71) | β = 0.06 [-0.19; 0.31] |
| Self-efficacy (n = 692) | | 8.51 (0.83) | 8.51 (1.08) | β = 0.04 [-0.27; 0.27] |
| Active choice (n = 743) | | 3.88 (0.52) | 3.80 (0.49) | β = 0.09* [0.01; 0.16] |
| Intention strength (n = 692) | | 8.32 (1.02) | 8.22 (1.10) | β = 0.10 [-0.20; 0.40] |
| Commitment (n = 692) | | 8.45 (0.93) | 8.13 (1.04) | β = 0.32* [0.04; 0.60] |

Note. Abbreviations: *CI* confidence interval, *SD* standard deviation, β regression coefficient, *OR* odds ratio, *CVD* cardiovascular disease

[a] A multinomial ordinal regression analysis compared the odds of the relevant option to the odds of 'no change' between the AC and PC group

[b] The results were log-transformed for the analysis using the natural logarithm

[c] This outcome was analysed using ordinal regression analysis

*$p < .05$.

**$p < .01$.

1.11;1.94). Response efficacy concerning lifestyle change was significantly higher in the AC than in the PC group (β = 0.24, 95%CI:0.06;0.42); however, groups did not differ in response efficacy concerning medication use (β = 0.06, 95%CI:-0.19;0.31). The AC group reported to have made a significantly more active choice than the PC group (β = 0.09, 95%CI:0.01;0.16). Intention strength and self-efficacy were high in both groups and did not significantly differ between groups; β = 0.10, 95%CI:-0.20;0.40 and β = 0.04, 95%CI:-0.27;0.27, respectively. Commitment to perform the chosen behavior was significantly higher in the AC group than in the PC group (β = 0.32, 95%CI:0.04;0.60).

**Results mediation analyses.** The results of our first mediation analysis are graphically shown in Fig 2. Cognitive risk perception and negative affect sequentially mediated the effect between the experimental condition (AC versus PC) and reported degree of active choice; indirect effect: β = 0.02; 95%CI: 0.01; 0.03. Given the total effect (β = 0.13, 95%CI: 0.06; 0.20), participants' cognitive risk perception and negative affect following this cognitive risk perception explained 17% of the relationship between the experimental condition and reported degree of active choice. Fig 3 shows the results of our second parallel mediation analysis. Reported degree of active choice explained 28% of the relationship between the experimental condition and commitment (indirect effect: β = 0.09, 95%CI: 0.02; 0.18; total effect: β = 0.32, 95%CI: 0.04; 0.60). However, autonomous motivation did not mediate the relationship

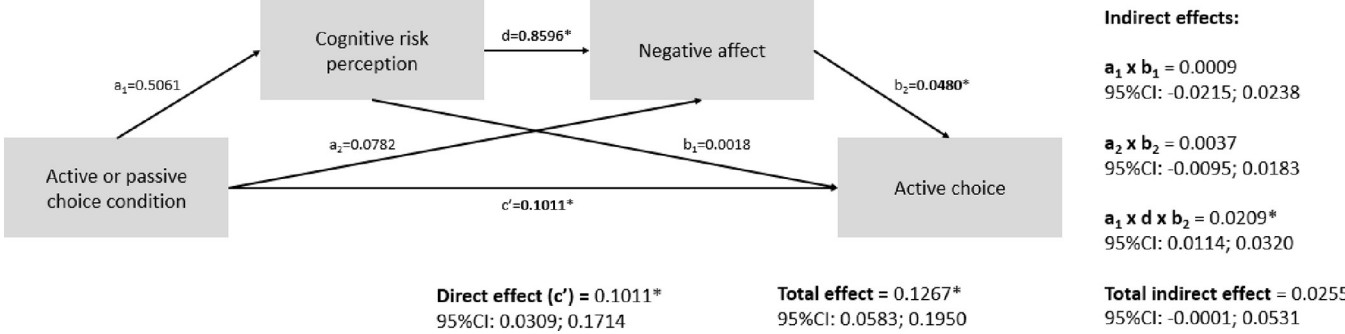

**Fig 2. A sequential mediation analysis shows the direct (f) and indirect paths by which experimental condition influences the degree of active choice.** Indirect, direct and total (direct + indirect) effects are reported along with their 95% confidence intervals. *p < .05.

between experimental condition and commitment (indirect effect: β = 0.09, 95%CI: -0.03; 0.23).

**Process evaluation.** Table 4 shows that the information provided to the AC and PC conditions was considered understandable, readable, well-organized, interesting, and educational. AC participants' evaluations were slightly more positive. Participants in both conditions indicated that information provided was not too much. AC participants rated 'information about the choice options' and 'information about the causes and consequences of CVD' as most helpful in decision-making. Table 5 presents AC and PC participants' suggestions for improvement.

## Discussion

The aim of this study was to find empirical support that fostering an active choice (AC), through an intervention based on risk awareness, leads to greater commitment and intention for behavioral change to reduce cardiovascular risk than information and advice only (i.e., a passive choice (PC)). In line with our expectations, the AC group reported to having made a more active choice and reported more commitment to perform the chosen behavioral change

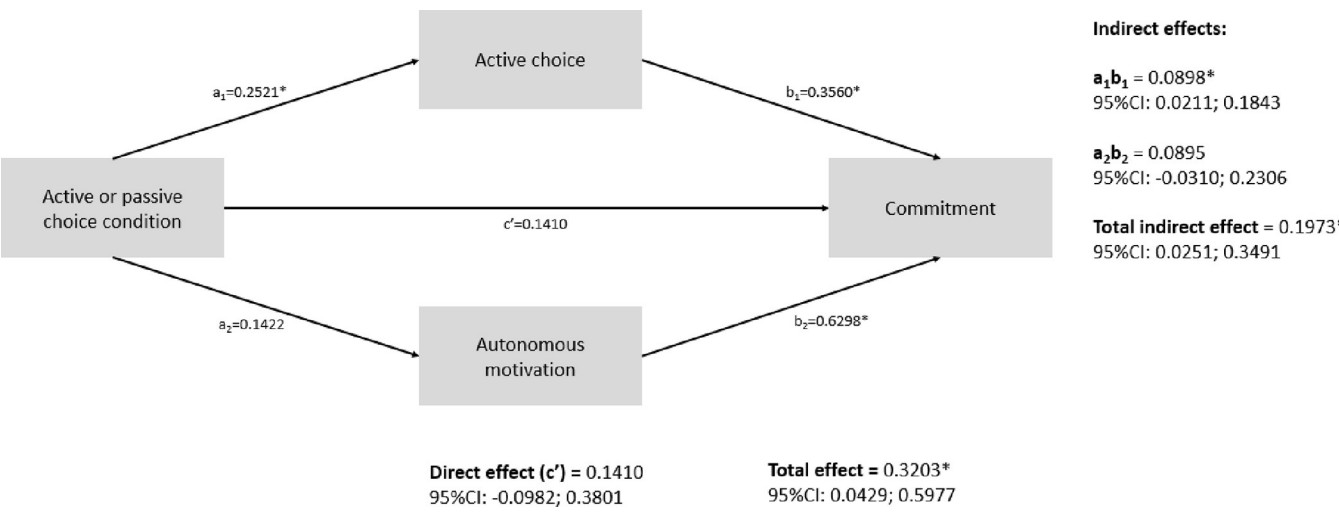

**Fig 3. A parallel mediation analysis shows the direct (c) and indirect paths by which experimental condition influences commitment.** Indirect, direct and total (direct + indirect) effects are reported along with their 95% confidence intervals. *p < .05.

**Table 4. Participants' general evaluation of the intervention (process evaluation).**

| Item[a] | Active Choice (AC) group | Passive Choice (PC) group |
|---|---|---|
| | Mean (SD) | Mean (SD) |
| *The following statements are about the information provided in this study. Please indicate to what extent you agree with the statements (scale: 1–5):* | | |
| The information was understandable to me | 4.27 (0.69) | 4.21 (0.62) |
| The information was readable to me | 4.36 (0.62) | 4.31 (0.54) |
| The information was well-organized | 4.27 (0.71) | 4.23 (0.61) |
| The information was interesting to me | 4.13 (0.76) | 3.97 (0.71) |
| I learned something from the information | 3.78 (0.93) | 3.59 (0.82) |
| It was too much information for me | 2.14 (0.82) | 2.08 (0.77) |
| *In this study, we asked you to consider lifestyle changes and medication use in case of an increased risk of cardiovascular disease. Next, you have chosen [selected option]. Please indicate for each component how well this has helped you in making that choice (scale: 1–5):* | | |
| The risk of CVD (i.e., heart age or 10-year absolute risk) | 3.87 (0.72) | 3.70 (0.67) |
| The information about the causes and consequences of CVD | 3.91 (0.59) | N/A |
| The information about the choice options (i.e., lifestyle change; medication use; lifestyle change and medication use; no change) | 3.92 (0.62) | N/A |
| The values-clarification exercise | 3.79 (0.79) | N/A |
| The information about lifestyle change and medication use | N/A | 3.86 (0.58) |
| The general practitioner's advice | N/A | 3.84 (0.59) |

Note. Abbreviations: *SD* standard deviation, *CVD* Cardiovascular disease

[a] All items were measured on a scale of 1 (totally disagree) to 5 (totally agree)

option than the PC group. As expected, the mediation analysis showed that AC participants' higher cognitive risk perception was associated with increased negative affect, which in turn was associated with a more active choice. The second mediation analysis revealed that reported degree of active choice–as a result of the AC intervention–positively contributed to commitment to the chosen behavioral change option, while there was no mediating relationship with autonomous motivation.

In choosing how to cope with increased CVD risk, AC participants reported having made a more active choice than PC participants. The finding that cognitive risk perception and negative affect sequentially mediated the relationship between experimental condition and active

**Table 5. Participants' suggestions for improvement.**

| AC participants | PC participants |
|---|---|
| Provide more concrete information. | Provide information about the pros and cons of lifestyle change and medication use to enable trade-offs. |
| Provide practical advice to support lifestyle change. | Present the risk reductions associated with different choice options. |
| Clarify how the heart age reduces if multiple behaviors are changed, as the presented reductions could not be added together. | Compare the CVD risk of 8% with the risk of others of the same age because the percentage in itself lacks meaning without a reference point. |
| Take existing healthy lifestyle habits into account. | |
| Use more images/icons. | |
| Provide more choice options (e.g., to first try one option to reduce CVD risk and only consider another option at a later time). | |

choice corresponds to the literature, which describes that cognitive risk perceptions can influence affect and, subsequently, behavior [38]. This finding indicates that risk awareness and a sense of urgency are essential for active decision-making, at least as measured with our self-constructed active choice measure. It must be noted, however, that cognitive risk perception and negative affect only partially mediated the effect on active choice. Other AC intervention components, which encouraged participants to evaluate pros and cons of choice options and consider their values, may have contributed to a more active choice as well.

In the current study as well as in one previous experiment [63], we found increased commitment to perform the chosen behavior following an active choice intervention. In line with expectations, the current study found that this relationship was mediated by reported degree of active choice. This corresponds to previous studies showing that autonomous decision-making and values-congruence, key aspects of an active choice, increase individuals' commitment and motivation [24, 29, 45]. Commitment has been associated with greater behavioral persistence, presumably because highly committed individuals have a lower chance to be affected by setbacks [22, 46, 47].

Contrary to expectations, AC participants did not report higher autonomous motivation than PC participants, and autonomous motivation did not mediate the relationship between condition (AC versus PC) and commitment. This finding might be related to the fact that the control group (PC) also reported relatively high autonomous motivation. AC and PC participants may both have perceived a high sense of autonomy because they were explicitly asked to choose the behavior they might want to change, which may have created explicit choice awareness. This might not correspond with a real-life situation in which risk-reducing advice is given by a GP. Alternatively, the scale to measure autonomous motivation–a subscale of the TSRQ–may not have been appropriate for our study, as we observed a ceiling effect in both groups. Ceiling effects for this subscale were also observed in other studies [e.g., 69, 70].

Intention strength and self-efficacy did not differ between the AC and PC groups; both reported relatively high intention strength and self-efficacy. This may be due to the hypothetical nature of our study. Concerning response efficacy, which together with self-efficacy determines individuals' coping response [61], we found that AC participants believed more strongly than PC participants that a healthy lifestyle would reduce CVD risk. However, AC and PC participants had less strong beliefs about the effectiveness of medication to reduce CVD risk. Previous research has demonstrated that patients can have significant concerns or skepticism about the value of preventive medication, for instance because of interested parties (e.g., academics, pharmaceutical industry) [71]. This might explain the lower response efficacy about medication in both AC and PC participants.

AC participants' increased cognitive risk perceptions, negative affect, and knowledge seem to be attributable to the heart age format and the accompanying information about CVD risk. Previous studies showed that a heart age format increases individuals' cognitive risk perception and negative affect compared to a 10-year absolute risk format [34, 37], probably because comparative information served as an anchor that facilitated cognitive and affective risk perception [34, 72]. AC participants' increased cognitive risk perceptions and negative affect were related to reported active choice and possibly explain why more AC than PC participants intended to reduce their risk by lifestyle changes or medication use [2, 37].

Our proof of concept study shows that fostering active choice among people at increased risk for cardiovascular disease may positively impact their intention and commitment for preventive action. This study provides empirical support for the assumption that fostering an active choice, through an intervention based on risk and choice awareness, increases commitment for CVD-risk reducing behavior. Further research in a real-world situation is needed to explore whether an active choice intervention results indeed in a higher percentage of

individuals choosing for risk-reducing behaviors. More research is also needed to study whether fostering active choice also results in greater behavioral persistence and less intentional and unintentional nonadherence in individuals at increased CVD risk. Based on our study, fostering active choice may be a valuable addition to motivate people at increased risk to change their behavior and adhere to a healthy lifestyle and a medication regimen.

### Strengths and limitations

Important strengths include the randomized controlled design and the theoretical foundations of the active choice intervention. Moreover, we presented risk reductions associated with lifestyle changes and medication use as reductions in heart age in the intervention, a relatively novel concept for making people aware of their CVD risk.

It should be acknowledged that many items were self-constructed, including the items to measure the degree of active choice, intentions, and commitment. Although we based this measure on a theoretically solid definition, pre-tested these items before data collection and calculated Cronbach's alpha values, the items and scales lack validity and evidence of psychometric properties. Another limitation is the observed ceiling effect for some variables, including autonomous motivation, which was assessed by a TSRQ subscale. Further, the performance of multiple statistical tests increases the likelihood of Type 1 errors [73]. We acknowledge the drawbacks of using a hypothetical risk scenario. Although hypothetical situations are not unusual in this research field [e.g., 34, 74], it can limit generalization of the results. Participants may have had trouble placing themselves in the hypothetical situation. Our study should therefore be considered as a proof-of-concept study, needing further studies in real-life situations.

### Conclusions

Using a hypothetical risk situation, our study provides evidence of the benefit of promoting an active choice to increase cognitive risk perceptions, negative affect, active decision-making, intention to take preventive actions, and stronger commitment towards CVD risk-reducing behavior. Active choice support regarding CVD risk management could be valuable for GPs to empower patients to make choices in line with their values.

### Declarations

**Trial registration.** ClinicalTrials.gov Identifier NCT05142280.

### Supporting information

**S1 File. SPSS research code main analyses.**
(PDF)

**S2 File. The Active Choice (AC) intervention.**
(PDF)

**S3 File. The Passive Choice (PC) control condition.**
(PDF)

**S4 File.**
(CSV)

## Acknowledgments

The authors would like to thank Annemieke Schaafstra & Tom Zandstra for sharing experiences regarding cardiovascular risk management in general practice; Jannick Dorresteijn and David Smeekes for sharing their thoughts on the calculation of heart age risk reduction; and Ehsan Motazedi for his advice on the statistical analyses.

## Author Contributions

**Conceptualization:** Lorraine L. Landais, Judith G. M. Jelsma, Olga C. Damman, Evert A. L. M. Verhagen, Danielle R. M. Timmermans.

**Data curation:** Lorraine L. Landais.

**Formal analysis:** Lorraine L. Landais.

**Funding acquisition:** Evert A. L. M. Verhagen, Danielle R. M. Timmermans.

**Investigation:** Lorraine L. Landais.

**Methodology:** Judith G. M. Jelsma, Danielle R. M. Timmermans.

**Project administration:** Danielle R. M. Timmermans.

**Supervision:** Judith G. M. Jelsma, Olga C. Damman, Evert A. L. M. Verhagen, Danielle R. M. Timmermans.

**Writing – original draft:** Lorraine L. Landais.

**Writing – review & editing:** Judith G. M. Jelsma, Olga C. Damman, Evert A. L. M. Verhagen, Danielle R. M. Timmermans.

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
