## [Decision Letter · Decision Letter 0]

13 Feb 2024

PONE-D-23-22195Fostering active choice to empower behavioral change to reduce cardiovascular risk: a web-based randomized controlled trialPLOS ONE

Dear Dr. Timmermans,

Thank you for submitting your manuscript to PLOS ONE. After careful consideration, we feel that it has merit but does not fully meet PLOS ONE’s publication criteria as it currently stands. Therefore, we invite you to submit a revised version of the manuscript that addresses the points raised during the review process.

We look forward to receiving your revised manuscript.

Kind regards,

Henri Tilga, PhD

Academic Editor

PLOS ONE

Journal Requirements:

   "Funded by Amsterdam University Medical Center" 

5. In the online submission form, you indicated that "The dataset used and analyzed during the current study is available from the corresponding author on reasonable request. Additional file 1 presents the code used for the statistical analyses."

6. Please include your tables as part of your main manuscript and remove the individual files. Please note that supplementary tables (should remain/ be uploaded) as separate "supporting information" files

Additional Editor Comments:

The Reviewer has provided several useful comments to increase the quality of this manuscript. Please carefully follow all the comments made by the Reviewer and revise the manuscript accordingly.

Reviewers' comments:

Reviewer's Responses to Questions

**Comments to the Author**

1. Is the manuscript technically sound, and do the data support the conclusions?

Reviewer #1: Yes

2. Has the statistical analysis been performed appropriately and rigorously? 

Reviewer #1: N/A

3. Have the authors made all data underlying the findings in their manuscript fully available?

Reviewer #1: Yes

4. Is the manuscript presented in an intelligible fashion and written in standard English?

Reviewer #1: Yes

5. Review Comments to the Author

Reviewer #1: PLOS One Review Comments to Author:

The paper presents the results of an experiment conducted with an online panel to examine the impact of active versus passive choice in decisions about cardiovascular disease prevention. The authors present a detailed background for the study and have strong theoretical grounding for the intervention and outcomes. The active choice intervention used a heart age risk presentation that had been previously shown to heighten risk perception. The findings point to the superiority of the active choice intervention over the passive choice condition. The paper is well written, yet there are a few areas that may benefit from clarification, mainly around the outcome measures and the results.

1. The primary outcome measure ‘active choice scale’ appears to be developed de novo for the study, as such, a bit more information about the development process would be helpful.

a. Please add more description regarding the constructs or components included, theoretical framework, and development process (e.g. cognitive testing of items, any validity testing/psychometric evaluation of the scale in addition to the Cronback alpha that is reported).

b. The authors report that they incorporated 5-items from an existing, well validated scale (Decisional Conflict Scale), why not just use that as its own measure as it has established benchmarks and strong psychometrics? Please comment on rationale for using subset of the Decisional Conflict Scale rather than the actual scale itself.

c. On a quick scan, the authors appear to have used active choice measures in other studies, e.g. Landais, L.L., Damman, O.C., Jelsma, J.G.M. et al. Promoting an active choice among physically inactive adults: a randomised web-based four-arm experiment. Int J Behav Nutr Phys Act 19, 49 (2022). https://doi.org/10.1186/s12966-022-01288-y . How is the 11-item measure used in this study similar to/different than the active choice measure employed in that study? Again, generally more information on this scale and development would be important to include as it is the primary outcome.

2. In Table 1, for all scales (e.g. active choice, knowledge, commitment) please provide the potential range for the total scores and then provide mean/SD/range for all of the outcomes measured.

3. Please clarify rationale for why self-efficacy and intention strength were measured with two items, yet only the highest score was used (which would seem to then exacerbate ceiling effects).

4. For commitment, two of three items were used to create a composite score. Again, the rationale is not clear as to why this was done, and why this was handled differently than self-efficacy and intention. Please provide rationale for the scoring of these and any psychometrics that are available for these items.

5. Study power – please comment on how the magnitude of active choice difference to power the study was selected and what that represents (is that clinically meaningful?) to help readers who are unfamiliar with this scale. It would also be helpful to report what that corresponds to in terms of standard deviation for the scale (e.g. 0.5 SD).

6. Table 3 – for cognitive risk perception, active choice and commitment, the differences appear quite small (though given comment #2, it is hard to know what the range of scores are for the variables). Are these differences meaningful?

7. After reading this a few times, it is still unclear how the ORs for the intentions variable are being calculated—is it within the arm or comparing across arms? While there does seem to be difference between arms in intention to pursue lifestyle changes (60% vs 67%), it is unclear how that translates to OR 2.86.

8. Discussion would benefit from reflection on the active choice measure (and possibly noting lack of validity or evidence of psychometric properties in the limitations).

6. PLOS authors have the option to publish the peer review history of their article (what does this mean?). If published, this will include your full peer review and any attached files.

Reviewer #1: No

---

## [Author Response · Author response to Decision Letter 0]

1 May 2024

We added the data file as a supplementary file. 

We added funding information 

Response to reviewer:

Point-by-point response to the reviewer

PONE-D-23-22195

We thank the editor and reviewer for carefully considering our manuscript. The comments and valuable suggestions of the reviewer helped us improve the manuscript. Below, we provide a point-by-point response to the comments.

Reviewer:

“The paper presents the results of an experiment conducted with an online panel to examine the impact of active versus passive choice in decisions about cardiovascular disease prevention. The authors present a detailed background for the study and have strong theoretical grounding for the intervention and outcomes. The active choice intervention used a heart age risk presentation that had been previously shown to heighten risk perception. The findings point to the superiority of the active choice intervention over the passive choice condition. The paper is well written, yet a few areas may benefit from clarification, mainly around the outcome measures and the results.”

1. The primary outcome measure ‘active choice scale’ appears to be developed de novo for the study, as such, a bit more information about the development process would be helpful.

a. Please add more description regarding the constructs or components included, theoretical framework, and development process (e.g. cognitive testing of items, any validity testing/psychometric evaluation of the scale in addition to the Cronback alpha that is reported).

Response: We agree with the reviewer that this requires further explanation. We developed the active choice scale to measure the degree of active choice. We did evaluate the suitability of existing instruments beforehand, including the Multi-dimensional measure of informed choice (Michie et al., 2002), However, it was discussed with the research team that these instruments were not sufficiently suitable for measuring the degree of active choice. Some questions from the Decisional Conflict Scale (specifically the 'Informed subscale' and 'Values clarity subscale') were deemed suitable for the active choice scale.

As there is no measure for active choice, we first defined the concept based on relevant literature (see introduction). We used this definition to develop our measure, as a solid theoretical foundation is essential for these kind of measures (see De Vet et al 2011). This scale was largely comparable to an active choice scale that we developed for a previous study (62). In accordance with the previous study, the first items of our active choice scale were adapted from the 'Informed subscale' and 'Values clarity subscale' of the validated Decisional Conflict Scale (63). The other subscales of the Decisional Conflict Scale were not included as they measured different constructs. Based on that study, the scale yields a reasonably normal data distribution. In the current study, we pre-tested the intervention and the questionnaire before data collection. One of the researchers (LL) used the 'think-aloud method': six adults went through the intervention and the questionnaire while reading everything aloud and verbalizing their thoughts aloud. This pre-test made some adjustments to the active choice scale, including clarifying the wording of some items. Our current active choice scale differs from the previous active choice scale regarding the wording of items: we adjusted the items to the current decision situation. Moreover, we added two items to the current scale to measure the extent to which participants had considered the consequences of (1) adopting a healthy lifestyle and (2) using medication for their risk of cardiovascular disease. Our active choice scale showed good internal consistency (α = 0.83). 

b. The authors report that they incorporated 5-items from an existing, well validated scale (Decisional Conflict Scale), why not just use that as its own measure as it has established benchmarks and strong psychometrics? Please comment on rationale for using subset of the Decisional Conflict Scale rather than the actual scale itself.

Response: We did not include the questions from the subscales 'support', 'uncertainty', and 'effective decision' of the Decisional Conflict Scale because they did not align with our construct of 'active choice'. We based our measure on a solid theoretical definition of active choice. The DCS is not a measure of active choice. 

c. On a quick scan, the authors appear to have used active choice measures in other studies, e.g. Landais, L.L., Damman, O.C., Jelsma, J.G.M. et al. Promoting an active choice among physically inactive adults: a randomised web-based four-arm experiment. Int J Behav Nutr Phys Act 19, 49 (2022). https://doi.org/10.1186/s12966-022-01288-y . How is the 11-item measure used in this study similar to/different than the active choice measure employed in that study? Again, generally more information on this scale and development would be important to include as it is the primary outcome.

Response: The active choice scale in our current study is comparable to the active choice scale used in our previous study. However, we adjusted the wording for the new decision situation since our current study was no longer about 'increasing physical activity' versus 'not increasing physical activity' but about adopting a healthier lifestyle or 'using medication'. Our previous active choice scale consisted of 9 items. To the current active choice scale, we added 2 items to measure the extent to which participants had considered the consequences of (1) adopting a healthy lifestyle and (2) using medication for their risk of cardiovascular disease.

Based on comments 1a-c, we made the following changes to the manuscript:

Lines 207-209: Before data collection, the questionnaire was also pre-tested among the six adults before data collection. Based on the pre-test, we made several adaptations to the questionnaires to increase comprehensibility.

Lines 227-243: To measure these outcomes, we used self-constructed items. These items were thoroughly evaluated with participants during the pretest and subsequently adjusted based on their experiences and feedback. The reported degree of active choice was measured using 11 self-constructed items (5-point scale). As there is no measure for active choice, we first defined the concept based on relevant literature (see introduction). We used this definition to develop our measure, as a solid theoretical foundation is essential for these kind of measures (see 62). This scale was largely comparable to an active choice scale that we developed for a previous study (63). In accordance with the previous study, the first items of our active choice scale were adapted from the 'Informed subscale' and 'Values clarity subscale' of the validated Decisional Conflict Scale (64). The other subscales of the Decisional Conflict Scale were not included as they measured different constructs. Based on that study, the scale yields a reasonably normal data distribution. In the current study, we pre-tested the intervention and the questionnaire before data collection. One of the researchers (LL) used the 'think-aloud method': six adults went through the intervention and the questionnaire while reading everything aloud and verbalizing their thoughts aloud. This pre-test made some adjustments to the active choice scale, including clarifying the wording of some items. Our current active choice scale differs from the previous active choice scale regarding the wording of items: we adjusted the items to the current decision situation. Moreover, we added two items to the current scale to measure the extent to which participants had considered the consequences of (1) adopting a healthy lifestyle and (2) using medication for their risk of cardiovascular disease. Our active choice scale showed good internal consistency (α = 0.83). 

2. In Table 1, for all scales (e.g. active choice, knowledge, commitment) please provide the potential range for the total scores and then provide mean/SD/range for all of the outcomes measured.

Response: We followed the reviewer's suggestion by adding this information to Table 1.

3. Please clarify rationale for why self-efficacy and intention strength were measured with two items, yet only the highest score was used (which would seem to then exacerbate ceiling effects).

Response: Initially, we planned to take the average of both scores on the items. However, we discovered this was not a logical starting point during the pre-test. For instance, if someone has a strong intention to make lifestyle changes (e.g., score 9) but a much weaker intention to use medication (e.g., score 2), then based on the average of those responses, one might wrongly conclude that the person does not have a strong intention to change behaviour. In contrast, they do, but only for one of the behaviours. Since we were interested in the choice option participants were most inclined to pursue, we only considered the highest score in the analyses of these variables. We explained this in the manuscript as follows.

Lines 224-225: This was done because we were interested in participants' confidence regarding the choice option they were most inclined to pursue.

Lines 249-250: In the analyses, we used the highest scores on these items since we were interested in the intention strength and commitment regarding the choice option participants were most inclined to pursue.

4. For commitment, two of three items were used to create a composite score. Again, the rationale is not clear as to why this was done, and why this was handled differently than self-efficacy and intention. Please provide rationale for the scoring of these and any psychometrics that are available for these items.

Response: We acknowledge that this was unclear. For commitment, we asked one question about the willingness to invest time, effort and energy into the chosen behaviour. In addition, we asked two questions about maintaining health behaviour: one about lifestyle and one about medication. For these last two questions, we again used the highest score (due to the reason above). Because we also had that first question about commitment, we calculated a composite score of this construct by taking the average of that first question and the highest score on those last two questions. Besides Cronbach's alpha value, we do not have any other psychometric information about this construct to report. In the manuscript, we added.

Lines 247-252: Concerning commitment to adopt risk-reducing behaviour, two items measured individuals’ commitment to maintaining behavioural changes on a 10-point scale: one about lifestyle change and one about medication use. In the analyses, we used the highest scores on these items since we were interested in the intention, strength and commitment regarding the choice option participants were most inclined to pursue. For commitment, a composite score was calculated by averaging the highest score on the two aforementioned items with the score on an additional item about the willingness to invest time, effort and energy into the chosen behaviour.

5. Study power – please comment on how the magnitude of active choice difference to power the study was selected and what that represents (is that clinically meaningful?) to help readers who are unfamiliar with this scale. It would also be helpful to report what that corresponds to in terms of standard deviation for the scale (e.g. 0.5 SD).

Response: We assumed an alpha of .05, a power of .80. We assumed a delta of 0.145 for active choice (because this lies exactly between 0.13 and 0.16; these were the significant differences found in an earlier study). We powered on active choice (based on only one group of the previous study: the control group) and then calculated the difference we can demonstrate with that calculated sample size on the measures intention strength and commitment. Since we found effect modification on several variables in the earlier study, e.g. intention strength and commitment, we powered by gender; that allows stratified analyzes for gender.

We added this information to the text:

Lines 276-277: “Since a previous study using an active choice intervention in a different setting (62) found that gender modified the effect on multiple outcomes, we stratified the current study by gender. We used SD=0.478 for men and SD=0.490 for women for the power calculation on ‘Active choice’ (62). We used a statistical power of 80%, an alpha level of 0.05, and a clinically relevant difference of 0.145 points on ‘Active choice’, meaning that 173 men and 182 women were needed per group. We assumed a delta of 0.145 for active choice because this lies exactly between 0.13 and 0.16; these were the significant differences found in an earlier study (62). Consequently, we could detect 0.74 points difference for men and 0.64 points difference for women on ‘Intention strength’, and 0.45 points difference for both men and women on ‘Commitment’.”

6. Table 3 – for cognitive risk perception, active choice and commitment, the differences appear quite small (though given comment #2, it is hard to know what the range of scores are for the variables). Are these differences meaningful?

Response: We agree that the differences are quite small. However, our study is a proof-of-concept study. We made this more clear in the current version of the paper. If our experiment shows an effect of an active choice intervention in an experiment however small, only then further studies are relevant how this could be tested in practice. 

Lines 120-123: This proof of concept study aimed to test the preliminary efficacy of fostering an active choice of participants at increased cardiovascular risk for preventive action as well as to better understand the mediating psychological processes using hypothetical scenarios. If our study shows positive results, the intervention could be tested in a real-life trial.

End discussion line 429-430: “Our proof of concept study shows that fostering active choice among people at increased risk for cardiovascular disease may positively impact their intention and commitment for preventive action. 

7. After reading this a few times, it is still unclear how the ORs for the intentions variable are being calculated—is it within the arm or comparing across arms? While there does seem to be difference between arms in intention to pursue lifestyle changes (60% vs 67%), it is unclear how that translates to OR 2.86.

Response: This intentions variable was analysed both within and across arms. That means that within each arm, the proportion of participants intending to pursue lifestyle changes was compared to those who intended to ‘change nothing’; these proportions were subsequently compared across the active choice and passive choice arms. This resulted in an odds ratio of 2.86. 

We added the following line to the manuscript to make this clearer:

Lines 330-331: For intention regarding coping with CVD risk, the results were compared both within and between study arms.

8. Discussion would benefit from reflection on the active choice measure (and possibly noting lack of validity or evidence of psychometric properties in the limitations).

Response: Our measure indeed lacks some psychometric testing. On the other hand, our measure has a solid theoretical base. Some concepts are based on items of the DCS which was shown to have good psychometric properties. We added a remark in the limitations section of the discussion.

Lines 443-446: It should be acknowledged that many items in the study were self-constructed, including part of the items to measure the degree of active choice, intentions, and commitment. Although we based this measure on a theoretically solid definition, pre-tested these items before data collection and calculated Cronbach’s alpha values, the items and scales lack validity and evidence of psychometric properties. 

Line 395 : The finding that cognitive risk perception and negative affect sequentially mediated the relationship between experimental condition and active choice corresponds to the literature, which describes that cognitive risk perceptions can influence affect and, subsequently, behavi

---

## [Editor Report · Decision Letter 1]

21 May 2024

Fostering active choice to empower behavioral change to reduce cardiovascular risk: a web-based randomized controlled trial

PONE-D-23-22195R1

Dear Dr. Timmermans,

We’re pleased to inform you that your manuscript has been judged scientifically suitable for publication and will be formally accepted for publication once it meets all outstanding technical requirements.

Kind regards,

Henri Tilga, PhD

Academic Editor

PLOS ONE

Additional Editor Comments (optional):

Authors have done well job on revising their manuscript.
---

## [Editor Report · Acceptance letter]

28 Jun 2024

PONE-D-23-22195R1 

PLOS ONE

Dear Dr. Timmermans, 

I'm pleased to inform you that your manuscript has been deemed suitable for publication in PLOS ONE. Congratulations! Your manuscript is now being handed over to our production team.

Kind regards, 

on behalf of

Dr. Henri Tilga 

Academic Editor

PLOS ONE